# Introducing HDAC-Targeting Radiopharmaceuticals for Glioblastoma Imaging and Therapy

**DOI:** 10.3390/ph16020227

**Published:** 2023-02-01

**Authors:** Liesbeth Everix, Elsie Neo Seane, Thomas Ebenhan, Ingeborg Goethals, Julie Bolcaen

**Affiliations:** 1Molecular Imaging Center Antwerp (MICA), University of Antwerp, 2610 Antwerpen, Belgium; 2Department of Medical Imaging and Therapeutic Sciences, Cape Peninsula University of Technology, Cape Town 7530, South Africa; 3Pre-Clinical Imaging Facility (PCIF), (NuMeRI) NPC, Pretoria 0001, South Africa; 4Department of Science and Technology/Preclinical Drug Development Platform (PCDDP), North West University, Potchefstroom 2520, South Africa; 5Nuclear Medicine, University of Pretoria, Pretoria 0001, South Africa; 6Department of Nuclear Medicine, Ghent University Hospital, 9000 Ghent, Belgium; 7Radiation Biophysics Division, SSC laboratory, iThemba LABS, Cape Town 7131, South Africa

**Keywords:** glioblastoma, histone deacetylases inhibitors, radiopharmaceuticals, theranostics

## Abstract

Despite recent advances in multimodality therapy for glioblastoma (GB) incorporating surgery, radiotherapy, chemotherapy and targeted therapy, the overall prognosis remains poor. One of the interesting targets for GB therapy is the histone deacetylase family (HDAC). Due to their pleiotropic effects on, e.g., DNA repair, cell proliferation, differentiation, apoptosis and cell cycle, HDAC inhibitors have gained a lot of attention in the last decade as anti-cancer agents. Despite their known underlying mechanism, their therapeutic activity is not well-defined. In this review, an extensive overview is given of the current status of HDAC inhibitors for GB therapy, followed by an overview of current HDAC-targeting radiopharmaceuticals. Imaging HDAC expression or activity could provide key insights regarding the role of HDAC enzymes in gliomagenesis, thus identifying patients likely to benefit from HDACi-targeted therapy.

## 1. Introduction

Glioblastoma multiforme (GB) is the most malignant tumor in the central nervous system (CNS). Despite recent advances in multimodality therapy for GB incorporating surgery, radiotherapy (RT), chemotherapy and targeted therapy, the overall prognosis remains poor. Almost all tumors recur with a more aggressive form, and there is no standard of care for recurrent GB. The survival rate at 5 years postdiagnosis remains at only 5.8% [1,2,3]. Novel molecular markers were identified improving GB classification and providing powerful prognostic information [4]. However, therapy resistance remains a hurdle. Precision oncology incorporating personalized targeted therapy holds much promise in developing more efficacious and tolerable therapies [3]. One of the interesting targets for GB-targeted therapy is the histone deacetylase family (HDAC). Due to their pleiotropic effects on, e.g., DNA repair, cell proliferation, differentiation, apoptosis and senescence, they have gained a lot of attention in the last decade as anti-cancer agents. In addition, HDAC inhibitors (HDACi) have been applied for the treatment of metabolic disorders and psychiatric or neurodegenerative diseases [5]. The HDAC family contains 18 family members, categorized as following: class I (HDAC1,2,3,8), IIa (HDAC 4,5,7,9), IIb (HDAC 6,10), III (nicotinamide adenine dinucleotide (NAD+)-dependent sirtuins (SIRTs) and IV (HDAC11) [6,7]. Two groups of enzymes control the acetylation and deacetylation of histones: histone acetyltransferase (HAT) and HDACs. The transfer or removal of acetyl groups by HATs and HDACs induce a more open and accessible chromatin structure or chromatin condensation and transcriptional repression [8,9]. Interestingly, the histone acetylation status is reversible and can be targeted by drugs [10]. HDACi have the ability to increase the level of protein acetylation in the cancerous cell, restarting the expression of silenced tumor suppressor genes [11]. However, despite their known underlying mechanism, their therapeutic activity is not well-defined.

The goal of this review is to highlight the current status of HDACi- and HDAC-targeting radiopharmaceuticals for the imaging and therapy of GB.

## 2. Role of HDAC in GB Pathology

Epigenetic mechanisms, particularly those involving enzymatic modifications of DNA and the associated histone proteins that regulate gene expression are recognized as a major factor contributing to the pathogenesis of GB [10]. The effects of acetylation on gene expression and tumor phenotype and the antitumor mechanism of HDACi in GB has recently been reviewed [11,12]. In gliomas, epigenetic enzymes, such as HDAC, are aberrantly expressed causing the deregulation of processes, featuring growth arrest, cell differentiation, cytotoxicity and apoptosis induction. RNA-sequencing data from the public TCGA (The Cancer Genome Atlas) showed that HDAC4, HDAC5, HDAC6, HDAC8 and HDAC11 expression was significantly lowered in glioma (WHO grade II–IV) when compared to normal brain tissue [9]. Expression levels of HDAC1-3 and HDAC7 appeared to increase with higher malignancy grades [9,13]. Additionally, HDAC3 and HDAC9 overexpression in GB are both correlated with a poor prognosis. The role of SIRT in GB is currently under debate [13].

In vitro, HDAC2 expression is significantly upregulated in GB [14]. The silencing of HDAC4 reactivated p21 (WAF1/Cip1) and inhibited tumor growth in an in vivo human GB model [15]. In addition, HDAC5 is upregulated in U87MG, U251MG, T98G and LN-229 glioma cell lines and promoted their proliferation by the upregulation of Notch 1 [16]. HDAC6 has been shown to promote the proliferation of glioma cells through the primary cilia, MKK7/JNK/c-Jun signaling pathway and attenuating transforming growth factor β (TGFβ) receptor signaling [17,18,19]. HDAC6 activity also plays a role in temozolomide (TMZ) resistance through the regulation of DNA mismatch repair [20].

HDACi are epigenome-targeting molecules divided into different categories based on their target and chemical structure: short-chain fatty acids (e.g., valproic acid (VPA)), hydroxamic acid derivatives (e.g., trichostatin A (TSA), vorinostat (SAHA), belinostat, panobinostat (LBH-589), pracinostat, quisinostat (JNJ-16241199)), carboxylic acid derivatives, cyclic peptides (romidepsin), and benzamides entinostat (MS-275), tacedinaline (CI-994) and mocetinostat (MG-0103)). Other categories include electrophilic ketones, hydro-examines, sirtuin inhibitors and miscellaneous [11,21,22]. FDA approval was granted for vorinostat (Zolinza^®^ Rahway, NJ, USA), belinostat (Beleodaq^®^, PXD101 East Windsor, NJ, USA), panobinostat (Farydak^®^ Barcelona, Spain) and romidepsin (Istodax^®^ Hayes, UK) for the treatment of hematological malignancies, represented by T-cell lymphomas and multiple myeloma (MM). Tucidinostat (Epidaza^®^, Chedamide, Shenzhen, China) was approved by China’s National Medical Products Administration [22]. Combined use of panobinostat with the proteasome inhibitor bortezomib has been approved for the treatment of refractory MM [6,7,21,23]. Advances in the abovementioned malignancies prompted HDACi-based anti-cancer research to expand to solid tumors, although the HDACi mechanisms of action in tumors are still sparsely understood. Figure 1 gives an overview of the confirmed mechanism of actions in GB, as recently reviewed [11,24]. As RT and TMZ therapy are standard in GB, the radiosensitizing and chemosensitizing effects of HDACi are of major interest. Presumably, HDACi promote a more open chromatin formation in tumor cells, thereby permitting DNA alkylating agents (e.g., TMZ) to access genomic DNA. Other mechanisms to reverse TMZ resistance have been suggested, e.g., blocking NF-κB-dependent transcription [13,14,25]. In particular, HDAC6 has been identified as a potential target for the treatment of TMZ-resistant GB [20,26,27]. The radiosensitization mechanism in GB could be induced by multiple mechanisms but eventually leads to a decrease in DNA repair [28,29,30,31,32,33,34,35]. Post-irradiation, HDACi have been shown to induce a prolonged expression of phosphorylated H2AX (γH2AX), a marker for DNA double strand breaks (DSBs) [33]. HDACi have also been shown to induce alterations in DNA replication, causing DNA damage [36,37]. Finally, HDACi may be able to assist in reversing abnormal genetic silencing, therefore leading to enhanced cell-cycle arrest and apoptosis from the action of DNA-damaging agents [13].

## 3. Current Status of HDACi for GB Therapy

An overview of the successful clinical trials investigating HDACi in high-grade glioma is given in Table 1. Studies in pediatric glioma patients were excluded. The previous reviews have focused on the mechanisms of HDACi in GB [11,38].

Most research reports on suberoylanilide hydroxamic acid (SAHA, vorinostat), a pan-HDACi, upregulating cancer suppressor genes (p21 (CDKN1A), PTEN, p27) and downregulating Akt-mTOR signaling, CDK2, CDK4 and cyclin D1/E. SAHA therapy triggered GB cell death and promoted hyper-radiosensitivity in wild-type p53 GB cells [10,34,39]. In GB patients, high doses of SAHA monotherapy appeared to be well-tolerated with modest single-agent activity. SAHA combination regimens with TMZ/RT and/or bevacizumab (BEV) have proven to be tolerable, but no statistical improvement in overall survival (OS) and/or progression-free survival (PFS) was noted [40,41,42,43,44,45]. Interestingly, phospholipase D1 (PLD1) has been identified as a target of resistance to vorinostat, and combined therapy with a PLD1 inhibitor might improve efficacy [46].

The hydroxymate-based pan-HDACi belinostat (PXD101, Beleodaq^®^) is structurally similar to SAHA but shows a greater blood brain barrier (BBB) passage [47]. In 2019, the potential of PXD101 was confirmed in an orthotopic rat glioma model. In a pilot study, PXD101 combined with TMZ/RT-delayed GB recurrence [47,48].

Depsipeptide romidepsin (Istodax^®^, FR901228, FK228) is a stable prodrug isolated from Chromobacterium violaceum and a class I HDACi [49,50]. In a phase I/II clinical trial in recurrent GB, romidepsin was found to be ineffective as a single agent [49].

Panobinostat (LBH589), a pan-deacetylase inhibitor of class I/II HDAC, is an antineoplastic and antiangiogenic drug that may work synergistically with BEV [51]. However, although this combined treatment strategy was well-tolerated, PFS and OS did not significantly improve compared to BEV monotherapy in recurrent GB [52]. A phase II trial is warranted to assess the combination with fractionated stereotactic re-irradiation therapy [53]. Panobinostat does not cross the BBB, and hence intratumoral or convection-enhanced delivery (CED) administration could be necessary [54].

HDACi valproic acid (valproate, VPA, Depakene), an anticonvulsive drug, has been shown to directly or synergistically exert inhibitory effects on glioma in vitro and in vivo [55]. VPA combined with TMZ/RT showed improvement in survival, but this might be limited to GB patients with wild-type p53 [56,57]. However, a phase III trial is warranted [58].

In the last 2 decades, an extensive amount of preclinical research on HDACi and multi-drug combinations in GB has been performed (see Appendix A) [19,33,68,69,70,71,72,73,74,75,76,77,78,79,80,81,82,83,84,85,86,87,88,89,90,91,92,93,94,95,96,97,98,99,100,101,102,103,104,105,106,107,108,109,110,111,112,113,114,115,116,117,118,119,120,121,122,123,124,125,126,127]. These studies provided new insights on HDACi-associated signaling processes.

HDACi appear to have a vital role in DNA damage response, and a radiosensitizing effect of HDACi (vorinostat, panobinostat, VPA, entinostat, scriptaid) has been shown in GB in vitro, with support for vorinostat for GB therapy in combination with heavy ion therapy [31,33,35,128]. HDACi have been shown to inhibit GB cell growth mediated by cell cycle arrest and apoptosis, as highlighted in Appendix A [129,130,131,132,133,134,135]. The class I/II HDACi trichostatin A (TSA) increased GB apoptosis induction through the p38MAPK-p53 cascade [136]. When combined with the proteasome inhibitor (MG132), 2-deoxy-d-glucose or lomustine (CCNU), synergistic apoptosis induction was shown [137,138,139]. The GB chemosensitization effects of HDACi therapy have also been noted, such as enhancement of TMZ-induced apoptosis [118,140,141,142,143]. However, vorinostat favored the evolution of TMZ resistance through O6-methylguanine DNA methyltransferase (MGMT) overexpression in GB in vivo [144]. HDACi SAHA and MC1568 blocked vascular mimicry in GB, and the inhibiting effects of HDACi on the invasiveness or migration of GB cells have been noted [143,145,146,147,148,149]. Multiple HDACi (vorinostat, romidepsin, MPT0B291, CDK4) have shown to increase the survival time of GB in vivo models [130,131,134,150,151].

Post-HDACi therapy, multiple genes that play a role in complex signaling pathways are up- or down-regulated, as recently summarized [11]. As expected based on preclinical data, the affected genes are involved in cell cycle progression, apoptosis, invasion and progrowth or include oncogenes and GSC markers [11]. 

Targeted drug combinations may beneficially affect the outcome of GB therapy, with the possible induction of synthetic lethality. Preclinically, promising combinations include a mix of epigenetic modifiers [152], HDACi combined with imipridones (activation of the mitochondrial ClpP protease) or proteasome inhibitors [153,154], panobinostat combined with a dual PI3K/mTOR inhibitor BEZ235 [155] and combining HDACi with MEK inhibitors or RTKi [156,157]. A triple combination therapy, involving panobinostat, OTX015 and sorafenib also showed potential in vitro [158]. Interestingly, the R132H mutation in isocitrate dehydrogenase 1 (IDH1R132H), commonly observed and associated with better survival in GB, has been linked to resistance to the anti-cancer effect of HDACi, such as TSA, vorinostat (SAHA) and valproic acid [159].

## 4. HDAC-Targeting Radiopharmaceuticals

The association of epigenetic dysfunction with disease and the development of diagnostic or therapeutic agents for treatment are challenging [160]. Most HDACi target a relatively wide spectrum of HDACs that, on their turn, inhibit various biological pathways. Their mechanisms of action as tumor suppressors have not yet been fully elucidated [10]. HDAC-targeting radiopharmaceuticals could provide better insights regarding HDAC tissue expression, HDACi biodistribution and pharmacokinetics and therapeutic efficacy and thereby unravel new insights into the function or behavior of HDACi in vivo [161,162]. Nuclear imaging of HDAC expression in GB may improve the understanding and roleplay of HDAC enzymes within gliomagenesis, identify patients likely to benefit from HDACi-targeted therapy and aid in optimizing therapeutic doses of novel HDACi for glioma treatment [163]. Importantly, there are two main strategies to consider when imaging an epigenetic target in the brain: 1) by direct observation (protein target information independent of its activity) and 2) functional observation (representative visualization of the impact of a protein or enzyme) [160]. Alternative methods to determine HDAC expression include invasive tumor biopsies and the use of peripheral lymphocytes as surrogate biomarkers for global acetylation after HDACi treatment.

An overview of HDACi-based radiopharmaceuticals is given in Figure 2, and Table 2 summarizes the preclinical development of HDAC radiopharmaceuticals. To visualize or treat GB with radiopharmaceuticals, it is particularly important to only consider those HDACi that sufficiently pass the BBB (even at sub-nanomolar concentration) and are of a small enough structure to allow their penetration into the bulky, heterogeneous tumor tissue [29]. In addition, the cellular location of the targeted HDAC needs to be considered, e.g., HDAC class I proteins are found predominantly in the nucleus, while class II proteins are primarily localized in the cytoplasm but can be shuttled between the cytoplasm and nucleus depending on their phosphorylation status [6].

To our knowledge, the potential of therapeutic HDAC radiopharmaceuticals for targeted radionuclide therapy (TRT) has not yet been explored. Importantly, possible brain toxicity may be a limiting aspect for this kind of application. HDACs play distinct physiological roles in the brain, and HDACi have pleiotropic effects due to their broad targets. This suggests a higher chance of success for isoform-specific HDACi or the necessity to inject such radioactive agents via CED directly into the GB tumor or its vicinity [164]. Another option is the use of nanovectors with theranostic properties to optimize the tumor delivery of potent HDACi, which could improve their anti-GB properties in vivo [165]. Other criteria to consider for the development of GB TRT agents were recently published by our group [166].

HDAC brain PET has been studied for the potential detection of various neurodegenerative diseases, such as Alzheimer’s and Parkinson’s disease, and limited studies have investigated their potential for glioma imaging [160,163,164]. Most HDAC radiopharmaceuticals are structurally related to SAHA and include 6-([^18^F]fluoroacetamido)-1-hexanoicanilide ([^18^F]FAHA), 6-(di-[^18^F]fluoroacetamido)-1-hexanoicanilide ([^18^F]DFAHA), 6-(tri-[^18^F]fluoroacetamido)-1-hexanoicanilide ([^18^F]TFAHA), [^18^F]F-SAHA (Figure 3A), *N*1-(4-(2-fluoroethyl)phenyl)-*N*8-hydroxyoctanediamide ([^18^F]FE-SAHA), [^18^F]fluoro-ethyltriazolesuberohydroxamine acid ([^18^F]FET-SAHA) (Figure 3B), [^125^/^131^I]-iodo-SAHA and two ^11^C-labeled SAHA-based ligands [162,167,168,169,170,171,172,173].

In 2006, the first ^18^F-labeled SAHA analogue ([^18^F]FAHA) was radiosynthesized by Mukhopadhyay et al. [168]. Soon thereafter, Nishii et al. confirmed PET in vivo brain uptake in rats of up to 0.44%ID/g between 5 and 60 min [169]. Moreover, blocking studies revealed a specificity similar to that of SAHA, suggesting that [^18^F]FAHA is a clinically relevant PET tracer capable of targeting HDAC IIa expression [170]. [^18^F]FAHA has also shown potential to monitor alterations in HDAC activity/expression in a rat model of chemotherapy-induced brain neurotoxicity [174]. Concerns were raised about [^18^F]fluoroacetate ([^18^F]FACE), a radiometabolite of the rapidly metabolized [^18^F]FAHA, that also crosses the BBB and therefore complicates [^18^F]FAHA quantification [171]. Fortunately, in non-human primates (NHP), the contribution of [^18^F]FACE to the ^18^F activity signal was minimal in the first 30 min post-administration [175]. A [^18^F]FAHA-like substrate developed by Seo et al. displayed an insufficient BBB permeability and HDAC specificity [167]. Bonomi et al. modified the structure of [^18^F]FAHA to add two or three fluorine groups ([^18^F]DFAHA or [^18^F]TFAHA, respectively), which increased the lipophilicity and thus BBB permeability [172]. In 2019, [^18^F]TFAHA was finally studied in GB rat models that confirmed tumor uptake 20 min post-radiotracer administration, which significantly reduced after administration of HDACi MC1568. [^18^F]TFAHA accumulation was also observed in normal brain structures known to overexpress HDAC class IIa: the hippocampus, nucleus accumbens, periaqueductal gray matter and cerebellum [163].

Brain uptake was reported of another radiolabeled SAHA-analogue, [^18^F]FE-SAHA, but its metabolic instability remains a substantial obstacle (high uptake in the kidneys, liver, bone and small intestines) [176]. Kim et al. developed [^18^F]FET-SAHA, which showed improved metabolic stability over [^18^F]FE-SAHA and accumulation in sarcoma tumors [173]. [^125/131^I]-iodo-SAHA maintained a comparable profile (e.g., similar toxicity and pharmacokinetics) to SAHA. However, in tumor-bearing mice, it showed no preferential tumor accumulation, rapid efflux and unspecific washout. Moreover, accumulation in the liver and kidneys was high [177]. Thus, none of the proposed SAHA-based radiopharmaceuticals have reached a clinical phase. 

Another group of HDACi-based radiopharmaceuticals, including [^11^C] trichostatin A, [^11^C]MS-275, [^11^C]KB631, [^11^C]4-phenylbutyric acid, [^11^C]valproate, [^11^C]butyric acid, [^11^C]CN89, [^11^C]CN107, [^18^F]F-panobinostat and [^11^C]PCI34051 demonstrated inadequate BBB penetration, which discourages their application for the HDAC-based imaging of GB, despite possible application in other tumor types [161,167,172,178,179,180,181]. [^18^F]F-panobinostat has bioactivity similar to that of unmodified panobinostat against diffuse intrinsic pontine glioma and U87MG glioma cells (nM efficacy), with low toxicity to healthy astrocytes [180]. [^18^F]F-panobinostat has also shown potential for PET-guided CED in order to achieve high brain concentrations in healthy mice and a pediatric diffuse midline glioma model, which could be translated to high-grade glioma (Figure 3C,D) [182]. 

Recently, radiolabeling of trifluoromethyloxadiazole (TFMO)-bearing class-IIa HDACi were explored, and NT160 was identified as a potent inhibitor of class-IIa HDAC4. [^18^F]F-NT160 was capable of BBB crossing, and binding to class-IIa HDACs was confirmed in mouse brain tissue [183].

In addition, radiometal-nuclide-labeled ligands have also been developed, such as a ^64^Cu-labeled hydroxamic acid-based radioligand 7-(4-(3-ethynylphenylamino)-7-methoxyquinazolin-6-yloxy)-N-hydroxyheptanamide (CUDC-101). CUDC-101 entered into Phase I clinical trial testing in multiple tumor types (but excluding glioma). [^64^Cu]Cu-CUDC-101 exhibits the capability to image HDAC expression in triple-negative breast cancer (Figure 3E–G). However, most radioligands conjugated to metal chelators fail to cross an intact BBB [184]. To this end, [^64^Cu]Cu-CUDC-101 may be a good candidate to explore CED-based administration for GB therapy.

As the development of highly brain-penetrant HDACi has been a persistent challenge, research is now shifting from radiolabeling existing HDACi to the development of novel brain-penetrant radiotracers; in particular, adamantane-conjugated radioligands seem promising [164]. The most advanced candidate is (E)-3-(4-((((3r,5r,7r)-adamantan-1-ylmethyl)([^11^C]methyl)amino)methyl)phenyl)-*N*-hydroxyacrylamide ([^11^C]martinostat), an adamantane-based hydroxamic acid with selective binding to HDAC 1, 2, 3 and 6 with subnanomolar potency and fast-binding kinetics [185]. In vivo, [^11^C]C-martinostat has shown a selective, reversible and dose-dependent binding, excellent signal-to-noise ratio and desirable safety profiles in rodents, pigs and humans [185,186,187]. Next, an ^18^F-labeled derivative of martinostat, [^18^F]F-MGS3, was developed by Strebl et al. [188]. [^18^F]F-MGS3 exhibited HDAC-specific binding, as well as comparable brain uptake and regional distribution compared to [^11^C]martinostat. However, [^18^F]F-MGS3 warrants more efficient radiosynthesis as poor yields and manual synthesis only allowed for low doses to be administered [188]. Lastly, [^18^F]F-bavarostat ([^18^F]F-EKZ-001) appears to be useful for HDAC6 quantification. In NHPs, [^18^F]F-EKZ-001 displayed rapid and high brain tissue uptake and excellent specific binding which was subsequently confirmed in healthy human adults [189,190]. 

**Table 2 pharmaceuticals-16-00227-t002:** Preclinical development of HDAC radiopharmaceuticals.

Radio- Pharmaceutical	(Pre)clinical Model	Year	Main Outcome and Findings	Ref
[^18^F]FAHA	Healthy rats	2007	(+) Uptake increased rapidly up to 0.44%ID/g (5–60 min). Target blocking (SAHA) decreased uptake	[169]
	Healthy NHP	2009	(−) Rapidly metabolized to [^18^F]FACE, which enters the brain	[171]
	Healthy NHP	2013	(-) Lack of BBB permeability and specificity	[167]
	Healthy NHP	2013	(+) BBB crossing. Limited influence of [^18^F]FACE to brain uptake (first 30 min)	[175]
	NNK-treated A/J mice	2014	(+) Midbrain, cerebellum and brainstem uptake was displaced by SAHA with <10% remaining	[170]
	Healthy mice	2018	(+) Specific uptake consistent with increased HDAC levels	[174]
[^18^F]DFAHA	Healthy rats	2015	(+) Selectivity for HDAC Class IIa > [^18^F]FAHA, favorably low unspecific brain accumulation	[172]
[^18^F]TFAHA	Healthy rats	2015	(+) Selectivity for HDAC class IIa > [^18^F]DFAHA and [^18^F]FAHA	[172]
	Intracerebral 9L and U87-MG rat xenografts	2019	(+) Increased accumulation at 20 min post-radiotracer administration (+) Target specificity, i.e., significant reduction uptake in 9L tumors after administration of HDACi MC1568 but not the SIRT1 specific inhibitor EX-527	[163]
	AD mouse model	2021	(+) Potential as an epigenetic radiotracer for AD	[191]
[^18^F]F-SAHA	A2780 OC mice	2011	(+) Exhibits nM potency. Target binding efficacy can be quantitated within 24 h	[162]
[^125/131^I]-iodo-SAHA	Thyroid, hepatoma, colon carcinoma- bearing mice	2008	(−) Equally toxic as SAHA. Rapid efflux and rapid washout and no preferential tumor accumulation. High (unwanted) accumulation in liver and kidneys	[177]
[^18^F]FE-SAHA	Mice LNCaP xenografts	2011	(+) Tumor uptake (−) High (unwanted) uptake in small intestines, kidneys, liver and bone (suspected defluorination)	[176]
[^18^F]FET-SAHA	RR1022 sarcoma rat	2018	(+) Significant accumulation in tumors with rapid blood clearance (gastrointestinal/renal excretion). Tracer accumulation was receptor-specific	[173]
[^11^C]TSA	Healthy NHP	2013	(−) Lack of BBB permeability and HDAC-specificity	[167]
[^11^C]MS-275	Healthy mice, rats, NHP	2010	(−) Poor brain penetration and lack of tracer specificity	[178]
[^11^C]KB631	B16.F10 murine melanoma-bearing mice	2019	(+) Showed HDAC6-selective binding (−) Lack of brain penetrance in rats, possibly due to the hydroxamate moiety	[181]
[^11^C]CN89	Healthy rats, NHP	2013	(−) Poor BBB penetration	[179]
[^11^C]CN107				
[^18^F]F-panobinostat	DIPG IV and XIII + U87MG glioma cells	2018	(+) Retains nM efficacy in glioma cells *in vitro*. Highly selective to glioma, with low toxicity to healthy astrocyte controls. Successful delivery to the murine central nervous system via CED (Figure 3D)	[180]
[^11^C]-4-phenylbutyric acid	Healthy NHP	2013	(−) Low brain uptake. Showed 15% metabolization after 30 min. High (unwanted) uptake in liver and heart	[161]
[^11^C]valproic acid	Healthy NHP	2013	(−) Low brain uptake. Showed 2% metabolization after 30 min. Exceptionally high (unwanted) heart uptake possibly due to its involvement in lipid metabolism	[161]
[^11^C]butyric Acid	Healthy NHP	2013	(−) Low brain uptake. Rapid metabolization (plasma: 80% metabolized after 5 min). Relatively high (unwanted) uptake in spleen and pancreas	[161]
[^11^C]PCI34051	Healthy rats/NHP	2013	(−) Poor BBB penetration. Low uptake in the brain within 80 min. Pretreatment with 2 mg/kg standard did not improve retention or permeability	[179]
[^64^Cu]Cu-CUDC-101	MDA-MB-231 xenograft mice	2013	(+) Specific binding to HDACs in vitro (nM). High TBR in vivo (Figure 3F)	[184]
[^11^C]martinostat	Healthy rats	2014	(+) Can quantify target engagement of structurally distinct, brain-penetrant hydroxamate HDACi in living rat brain	[192]
	Healthy NHP	2014	(+) Highly selective and specific. Testing in humans is warranted	[185]
	Healthy NHP	2015	(+) Allows quantification of brain HDAC expression. Reversible and dose-dependent binding. Slow washout kinetics observed	[193]
	Healthy humans	2016	(+) Selectively binds HDAC1, 2 and 3	[186]
	Healthy pigs	2020	(+) Allowed for accurate in vivo measurement of cerebral HDAC1—3 protein levels. Excellent signal-to-noise ratio	[187]
[^18^F]F-MGS3	Healthy rats, NHP	2016	(+) Exhibits specific binding/comparable brain uptake and regional distribution to [^11^C]martinostat	[188]
[^18^F]F-bavarostat	Healthy rats, NHP	2017	(+) Selective HDAC6 inhibitor. Excellent brain penetrance. Low amount of nonspecific binding observed after pre-treatment with 1 mg/kg unlabeled bavarostat	[194]
	Healthy NHP	2020	(+) Excellent brain penetrance. Good HDAC6 selectivity, enabling quantification	[189]
	Healthy humans	2021	(+) Safe to administer and accurate quantification of HDAC6 expression in the human brain	[190]
[^18^F]F-NT160	Healthy mice	2022	(+) Can cross the BBB and bind to class-IIa HDACs in vivo in mice brain tissue	[183]

AD = Alzheimer’s disease; BBB = blood brain barrier; CED = convection enhanced delivery, CNS = central nervous system; DIPG = diffuse intrinsic pontine glioma; HDAC = histone deacetylase; i.v. = intravenous; LNCaP = lymph node carcinoma of the prostate; NHP = non-human primate; NNK = 4-(methylnitrosamino)-1-(3-pyridyl)-1-butanone; OC = ovarian cancer; PET = positron emission tomography; TBR = tumor-to-background ratio.

## 5. Challenges and Future Outlook

Although extensive research has been performed on HDACi in glioma with clear radio- and/or chemosensitizing effects, the potential of radiolabeled HDACi has only been confirmed in the field of neurodegenerative diseases and been primarily diagnostic, with the goal of quantifying HDAC expression and/or monitoring treatment response. Their potential for GB imaging and TRT is underexplored. Whilst furthering this field of research should be recommended, one of the major issues that slowed down recent translation to clinics was poor BBB penetration, poor specificity and diverse target locations. Interestingly, adamantane-conjugated radioligands seem promising to increase brain penetrance [164]. Only two radiotracers have been investigated in healthy adults: [^11^C]martinostat and [^18^F]F-bavarostat, confirming the ability to quantify HDAC expression [186,190]. Both should be recommended for GB HDAC imaging as they have shown target specificity and reported brain penetrance. [^18^F]TFAHA is the only radiopharmaceutical that has been evaluated in GB models, with uptake in GB tumors but also in normal brain structures known to overexpress HDAC class IIa [163]. Another recommendable radiopharmaceutical is [^18^F]F-NT160 featuring potent binding to class-IIa HDACs and BBB crossing in mice [183]. However, future studies are needed to increase its tumor specific uptake while preventing damage to healthy tissues.

The potential for HDACi-based radiopharmaceuticals in GB can currently be formulated as (1) biomarkers for HDAC expression, (2) elucidate the roles of HDAC class enzymes and (3) dose optimization of cold HDACi [163]. Cancer resistance and the toxic effects of HDACi are currently an issue to translate radiolabeled HDACi for potential application in TRT. HDACi are often pan-specific towards a specific HDAC class. As their substrates are present all over the human brain, targeting HDACi in GB may cause unwanted effects on healthy tissues too. However, CED could be considered to mitigate any adverse effects and circumvent the BBB. Targeting multiple HDAC proteins could also be advantageous due to the heterogeneous nature of GB. Research should be initiated to confirm this, including optimal combinatorial strategies for HDACi that permit efficacy as well as safety in GB.

## Figures and Tables

**Figure 1 pharmaceuticals-16-00227-f001:**
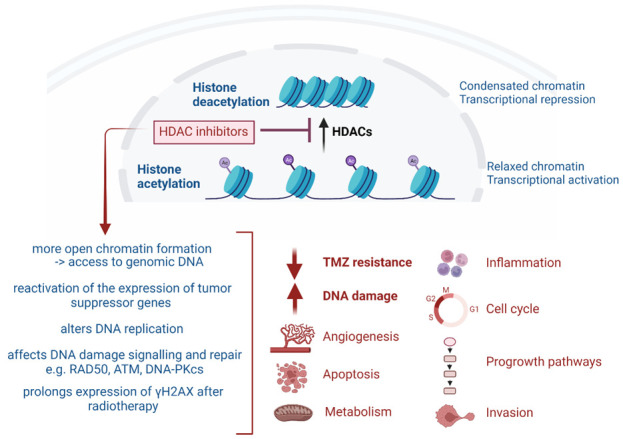
Overview of the broad effects of HDAC inhibitors.

**Figure 2 pharmaceuticals-16-00227-f002:**
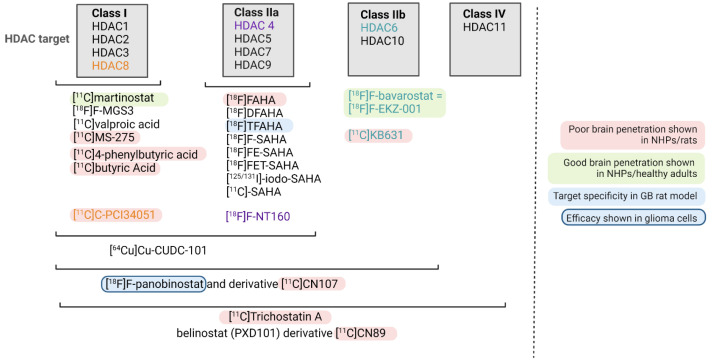
Overview of current radiopharmaceuticals targeting HDAC. The HDAC class targeted is shown. If only one HDAC enzyme is targeted, this was highlighted in color (orange, purple and green).

**Figure 3 pharmaceuticals-16-00227-f003:**
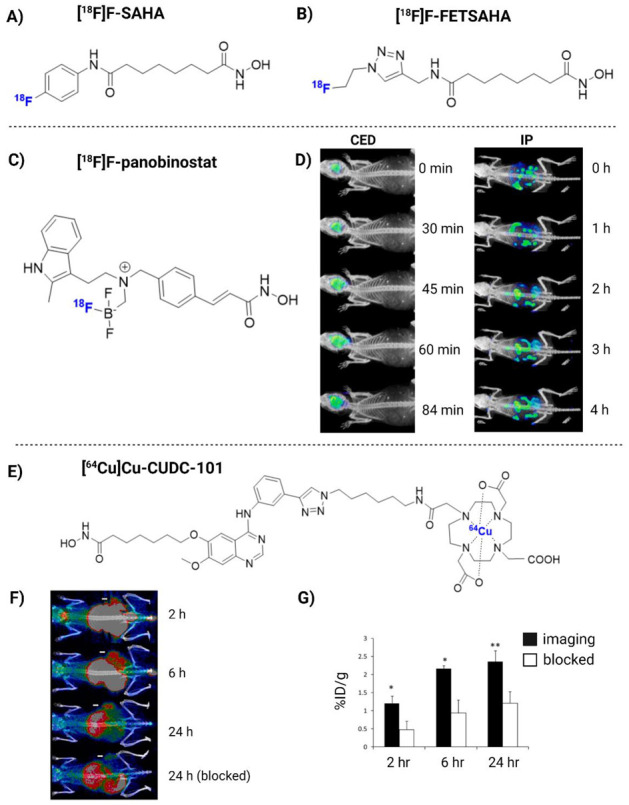
The chemical structure of [^18^F]F-SAHA (**A**), [^18^F]FET-SAHA (**B**), [^18^F]F-panobinostat (**C**) and [^64^Cu]Cu-CUDC-101 (**E**). [^18^F]F-panobinostat PET imaging when delivered via convection-enhanced delivery (CED) or intraperitoneal (IP) (**D**). In vivo PET imaging of [^64^Cu]Cu-CUDC-101 in MDA-MB-231-bearing tumor mice, with or without co-injection of cold CUDC-101 (**F**,**G**). Images reproduced with permission from [180,184]. Copyright 2013 and 2018 American Chemical Society.

**Table 1 pharmaceuticals-16-00227-t001:** Overview of clinical trials on HDAC inhibitors (HDACi) in high-grade glioma.

		Regimen	Stage	GB Type	Main Result	Reference
Vorinostat (SAHA, Zolinza, MK0683) *Pan-HDACi*	(+)	/	C(II)		Well-tolerated. Modest single-agent activity. Trials with combination regimens warranted	[40]
		TMZ	NA	rec/prog	MRS imaging may enable quantitative analysis of tumor response	[59] NCT01342757 *
		TMZ	C(I)	HGG	Well-tolerated	[41]
		BEV/Irinotecan	C(I)	rec	(+) Well-tolerated (+) OS and PFS at 400 mg daily or 300 mg twice a day	NCT00762255 *
	(−)	BEV	C(II)	rec	PFS6 or median OS was not improved	[45] NCT01738646 *
		BEV	C(I/II)	rec	Did not improve PFS or OS	[43] NCT01266031 *
		BEV/TMZ	C(I/II)		PFS6 was not statistically improved beyond controls	[44] NCT00939991 *
		BEV/CPT-11	C(I)	rec	Increased toxicities	[60]
		Erlotinib/BEV	C(I/II)	rec	Trial terminated (toxicities)	NCT01110876 *
		Bortezomib	C(II)	rec	Trial closed at interim analysis (0/34 progression-free)	[61] NCT00641706 *
		FSRT	C(I)	rec	Trial terminated	NCT01378481 *
		TMZ/RT	C(I/II)	nd	Acceptable tolerability, but primary efficacy endpoint not met. Sensitivity signatures could facilitate patient selection	[62] NCT00731731 *
	Ongoing	TMZ	C(I)	HGG	Active: not recruiting	NCT00268385 *
		TMZ/Carboplatin/Isotretinoin	C(I/II)	rec	Active: not recruiting	NCT00555399 *
		Pembrolizumab/TMZ/RT	C(I)	nd	Active: recruiting	NCT03426891 *
Belinostat (PXD101, Beleodaq) *Pan-HDACi*	Ongoing	TMZ/RT	Pilot study	nd	Active: not recruiting Radiosensitizing effect	[47,48] NCT02137759 *
Romidepsin (Istodax, FK228, FR901228, depsipeptide) *HDAC class I*	(−)		C(I/II)	rec	Ineffective	[49] NCT00085540 *
	Ongoing		C(I)	glioma	Active: not recruiting	NCT01638533 *
Panobinostat (LBH589) *HDAC class I/II*	(+)	FSRT	C(I)	recHGG	Well-tolerated. Phase II trial warranted	[53]
	(−)	BEV	C(II)	recHGG	Did not improve PFS6 compared to BEV monotherapy	[52] NCT00859222 *
			C(II)	recHGG	Trial terminated due to insufficient accrual	NCT00848523 *
			C(II)	rec	Trial withdrawn due to no enrollment	NCT01115036 *
Valproic acid (VPA, valproate, Depakene) *HDAC class I*	(+)		C(II)	HGG	Well-tolerated. May result in improved outcomes. A phase III should follow	[56] NCT00302159 *
				HGG	Delayed hair loss and improvement in survival	[63]
				HGG	Improvement in survival	[64]
				nd	Improvement in PFS and OS confirmed	[58]
				nd	Survival benefit dependent on their p53 gene status	[57]
		Levetiracetam/TMZ/RT	Retro	GB	2-months longer survival	[65]
	(−)	Doxorubicin/TMZ/RT	C(II)	nd	Trials terminated	NCT02758366 *
		Celecoxib	C(II)	Nd		NCT00068770 *
		SRS/Nivolumab	C(I)	Rec		NCT02648633 *
			psa	GB	PFS and OS were comparable to historical controls	[66]
		TMZ	Retro	II/III	VPA was linked to histological progression and decrease in PFS	[67]
	Ongoing	Sildenafil/Sorafenib	C(II)	recHGG	Active: not recruiting	NCT01817751 *
		Levetiracetam	C(IV)	glioma	Recruiting: for seizure treatment	NCT03048084 *
		Perampanel	C(IV)	HGG	Recruiting: for seizure treatment	NCT04650204 *
		TMZ	C(III)	HGG	Recruiting	NCT03243461 *

/ = monotherapy. * For current state of clinical trials, visit ‘https://clinicaltrials.gov/’ (accessed on 1 October 2022) [42]. BEV = bevacizumab; FSRT = fractionated stereotactic re-irradiation therapy; GB = glioblastoma; HGG = high-grade glioma; MRS = magnetic resonance spectroscopy; NA = not applicable; nd = newly diagnosed glioblastoma; OS = overall survival; PFS6 = progression-free survival (6 months); prog = progressive glioblastoma; psa = prospective single-arm study; rec = recurrent glioblastoma; Retro = retrospective; RT = radiotherapy; SRS = stereotactic radiosurgery; TMZ = temozolomide. Note: trials in pediatric glioma patients were not included.

## Data Availability

Not applicable.

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
