# Peer review of "Introducing HDAC-Targeting Radiopharmaceuticals for Glioblastoma Imaging and Therapy"

_pharmaceuticals, 2023, doi:10.3390/ph16020227_

Round 1

Reviewer 1 Report

Please, check spelling.

Author Response

Dear reviewer,

We appreciate your positive review. 

Thanks for noticing some spelling errors. We did a thorough spell check and corrected all mistakes on line 111, 113, 116, 125, 199, 201, 202, 203, 211, 213, 224, 249, 257, 267, 305, 330.

Based on the comments of other reviewers we added Figure 3 and referred to it on line 227, 229, 268, 278, 305.

Note that Thomas Ebenhan made some corrections to his affiliations and a graphical abstract was submitted.

Reviewer 2 Report

This review article from Everix et al offers an overview of the current status of research about histone deacetylase (HDAC) inhibitors for glioblastoma therapy, as well as HDAC targeting radiopharmaceuticals for GB imaging. A comprehensive view of this field of research is reported, with very useful tables summarizing the most relevant publications in both therapy and radionuclide imaging. To the best of the knowledge of this reviewer, the work is original; morover, it appears well structured and a good reference for all investigators involved in GB therapy and diagnosis research. Also, the supplemental materials provide even a more comprehensive list of literature resources, complementing those reported in the main manuscript. Overall, I feel that this form of the manuscript is almost ready for publication.

I just noticed few typos in the manuscript:
1) Line 123: "BBB". Blood brain barrier has been defined only in the caption of Table 3 at page 10, but it must be written in extenso also at its first occurrence in the main text.
2) Line 196: "vizualize" -> visualize;
3) Line 207: "pleiotropc" -> pleiotropic;
4) Line 209: "ots" -> its;
5) Lines 253-254: "have radiopharmaceuticals have reached a clinical phase". This sentence must be corrected.
6) Line 320: "HIDAC" -> HDAC

Author Response

Dear reviewer,

We appreciate your positive review. 

Thanks for noticing some spelling errors. We did a thorough spell check and corrected all mistakes on line 111, 113, 116, 125, 199, 201, 202, 203, 211, 213, 224, 249, 257, 267, 305, 330.

Based on the comments of other reviewers we added Figure 3 and referred to it on line 227, 229, 268, 278, 305.

Note also that Thomas Ebenhan made some corrections to his affiliations and a graphical abstract was submitted.

Reviewer 3 Report

This review article summarizes the recent development of HDAC targeting inhibitors for GBM treatment and radiopharmaceuticals. Overall, the paper is interesting and informative. It can be accepted for publication after addressing the following minor concerns.

1)      It would be helpful to add a Figure shows the chemical structure of representative HDAC targeting inhibitors and radiopharmaceuticals.

2)      It would be helpful to add a Figure showing some of the imaging results of the radiopharmaceuticals developed for different models.
